# A comparison of the Netherlands, Norway and UK familial hypercholesterolemia screening programmes with implications for target setting and the UK's NHS long term plan

Christopher Page[1], Huiru Zheng[2], Haiying Wang[2], Taranjit Singh Rai[3], Maurice O'Kane[4], Pádraig Hart[5], Shane McKee[5], Steven Watterson[3]*

1 Personalised Medicine Centre, School of Biomedical Science, Ulster University, C-TRIC Building, Altnagelvin Area Hospital, Derry, Northern Ireland, United Kingdom, 2 School of Computing, Ulster University, Belfast, Northern Ireland, United Kingdom, 3 Personalised Medicine Centre, School of Medicine, Ulster University, Derry, Northern Ireland, United Kingdom, 4 Western Health and Social Care Trust, Altnagelvin Area Hospital, Derry, Northern Ireland, United Kingdom, 5 Northern Ireland Regional Genetics Service, Belfast City Hospital, Belfast, Northern Ireland, United Kingdom

* s.watterson@ulster.ac.uk

## Abstract

We sought to determine the most efficacious and cost-effective strategy to follow when developing a national screening programme by comparing and contrasting the national screening programmes of Norway, the Netherlands and the UK. Comparing the detection rates and screening profiles between the Netherlands, Norway, the UK and constituent nations (England, Northern Ireland, Scotland and Wales) it is clear that maximising the number of relatives screened per index case leads to identification of the greatest proportion of an FH population. The UK has stated targets to detect 25% of the population of England with FH across the 5 years to 2024 with the NHS Long Term Plan. However, this is grossly unrealistic and, based on pre-pandemic rates, will only be reached in the year 2096. We also modelled the efficacy and cost-effectiveness of two screening strategies: 1) Universal screening of 1-2-year-olds, 2) electronic healthcare record screening, in both cases coupled to reverse cascade screening. We found that index case detection from electronic healthcare records was 56% more efficacious than universal screening and, depending on the cascade screening rate of success, 36%-43% more cost-effective per FH case detected. The UK is currently trialling universal screening of 1–2-year-olds to contribute to national FH detection targets. Our modelling suggests that this is not the most efficacious or cost-effective strategy to follow. For countries looking to develop national FH programmes, screening of electronic healthcare records, coupled to successful cascade screening to blood relatives is likely to be a preferable strategy to follow.

**Data Availability Statement:** All data is contained within the paper.

**Funding:** This work was supported by funding from the Department for the Economy, Northern Ireland, to all authors. It was also supported by funding from the Public Health Agency (PHA) of the Health and Social Care (HSC) R&D Division and the the Western Health & Social Care Trust (to TSR with award number COM/5618/20). The funders had no role in study design, data collection and analysis, decision to publish, or preparation of the manuscript.

**Competing interests:** The authors have declared that no competing interests exist.

# 1. Introduction

Familial hypercholesterolaemia (FH) is a common genetic disorder with a prevalence of approximately 1 in 250 of the general population [1]. FH affects lipoprotein metabolism [2, 3] and can drive atherosclerosis [4, 5]. It is characterised by elevated serum low-density lipoprotein cholesterol (LDL-C), usually as a result of compromised catabolism of circulating LDL-C [3]. Worldwide, a birth occurs with heterozygous familial hypercholesterolemia (HeFH) approximately every minute and with homozygous familial hypercholesterolaemia (HoFH) approximately every day [6].

Prior to the availability of statins [7], a nine-year follow-up study of a UK cohort identified adults aged 20–39 years as the age-bracket with the highest ratio of observed to expected deaths, with a standardised mortality ratio (SMR) of 96.86 in contrast to adults reaching 40–59 years of age with an SMR of 3.86 [8]. Similarly, a follow-up study of a Norwegian cohort identified adults aged 20–39 years as the age-bracket with the highest SMR [9]. With the introduction of statins, the SMR for statin-treated adults with FH aged 20–79 years fell to 2.4 for 'definite FH' and 1.78 for 'possible FH' [10]. FH has been widely recognised to fulfil the Wilson and Jungner criteria to justify a screening programme, due to its importance, the quality of our understanding and the cost-efficacy of diagnosis and treatment [11–13]. However, it is estimated that >92% of individuals with FH in the UK remain undetected and therefore untreated [14].

The NHS Long Term Plan (LTP) was published in January 2019 as a strategic plan to improve the detection and provision of care for individuals with heart, circulatory and respiratory diseases in England [15]. The NHS LTP recognises heart & circulatory diseases as having the greatest potential to save lives over the next 10 years. This plan aims to prevent up to 150,000 heart attacks, strokes, and reduce risks of dementia over the period of 2019–2029 [15]. In particular, the LTP specifically prioritises accessibility to genetic testing and treatment for FH, setting the target of identifying 25% of FH positive individuals within 5 years, from 2019 to 2024 [15]. However, in England there has historically been little in the way of coordinated national screening and the LTP targets represent a contribution to its long-term development. Here, we compare and contrast the historical screening efforts of the UK to those of Norway and the Netherlands, countries with well-established screening programmes [16, 17], and consider what can be learned regarding target setting and the development of screening programmes.

# 2. Methods

## 2.1 Screening rate estimates

The number of individuals identified with FH for the Netherlands, Norway, England, Scotland, Wales, Northern Ireland and the UK as a whole were sourced from published reports on cascade testing (see Table 1). Start years for FH screening were determined from published reports with Norway screening since 1998 [18] and the Netherlands since 1994 [19]. Genetic testing has been available in Northern Ireland since 2000 [20]. Screening in England is reported to have started in 2003 [21]. Testing in Wales and Scotland commenced in 2005 and 2008, respectively [21]. In our analysis, we estimate detection rates to compare UK screening to Norway and the Netherlands.

The total FH population was determined by multiplying the putative prevalence of FH by country- and year-matched population sizes. Population sizes for England, Scotland, and Wales were sourced from the Office for National Statistics [30] and the Northern Ireland Statistics and Research Agency [31]. The Netherlands and Norway's population sizes were sourced from World Bank population statistics [32, 33]. The country-specific prevalence of FH

**Table 1. Numbers and proportions of estimated FH populations reported in the published literature.** The most rapid screening programmes belong to Norway and Netherlands, both countries with national screening efforts, generally identifying >1% of the FH population per year. The UK and constituents follow a slower trajectory, generally identifying <1% of the FH population per year.

| Region | Genetic testing since (year) | Study | Year reported | Population measured (M) | FH population size | FH population identified | % FH population identified | Detection rate since start (% per year) | Detection rate in report period (% per year) |
|---|---|---|---|---|---|---|---|---|---|
| Norway | 1998 | Leren, 2004 [18] | 2003 | 4.57 | 14,600 | 407 | 2.79 | 0.56 | |
| | | Mundal, 2014 [22] | 2010 | 4.89 | 15,623 | 4,688 | 30.00 | 2.5 | 3.89 |
| | | Mundal, 2017 [23] | 2013 | 5.08 | 16,230 | 5,518 | 34.00 | 2.27 | 1.33 |
| | | Leren & Bogsrud, 2021 [24] | 2020 | 5.38 | 17,188 | 8,811 | 51.26 | 2.33 | 2.47 |
| The Netherlands | 1994 | Umas-Eckenhausen, 2001 [19] | 1999 | 15.81 | 63,240 | 2,039 | 3.22 | 0.64 | |
| | | Fouchier 2001 [25] | 2001 | 16.05 | 63,200 | 5,531 | 8.75 | 1.25 | 2.76 |
| | | Fouchier 2005 [26] | 2004 | 16.28 | 65,120 | 9,897 | 15.20 | 1.52 | 2.15 |
| | | Kusters, 2011 [27] | 2009 | 16.53 | 66,120 | 14,805 | 22.39 | 1.49 | 1.44 |
| | | Louter, 2017 [28] | 2014 | 16.87 | 67,480 | 28,000 | 41.49 | 2.07 | 3.82 |
| Northern Ireland | 2000 | Cather, 2016 [20] | 2016 | 1.86 | 7,440 | 1,034 | 13.90 | 0.87 | |
| | | Kerr, 2017 [29] | 2017 | 1.86 | 7,480 | 1,095 | 14.64 | 0.86 | 0.74 |
| | | Haralambos, 2018 [21] | 2018 | 1.87 | 7,480 | 1,256 | 16.79 | 0.93 | 2.24 |
| Scotland | 2008 | Haralambos, 2018 [21] | 2018 | 5.37 | 21,480 | 1,825 | 8.50 | 0.85 | |
| Wales | 2005 | Haralambos, 2018 [21] | 2018 | 3.17 | 12,680 | 1,000 | 7.89 | 0.61 | |
| United Kingdom | 2000 | Haralambos, 2018 [21] | 2018 | 29.8 | 119,200 | 7,168 | 6.01 | 0.33 | |
| England | 2003 | Haralambos, 2018 [21] | 2018 | 19.4 | 77,600 | 3,087 | 3.98 | 0.27 | |
| Scotland, Wales, Wessex (pooled data) | 2003 | Kerr, 2017 [29] | 2017 | 21.4 | 85,600 | 2,466 | 2.88 | 0.21 | |

was determined from the most recent published reports on the identification of individuals with FH through cascade testing. This yielded a prevalence of 1 in 313 for Norway [24] and 1 in 250 for the Netherlands [28]. For the UK, the prevalence was assumed to be 1 in 250 [1].

The number of FH cases detected as index cases and as relatives was reported directly in all studies. In most studies, the number of genetic tests was also reported. However, where it was not reported, we have estimated the value on the assumption that relatives detected were parents/children/siblings of heterozygous cases and therefore likely to share the FH genotype with a probability of 0.5. We model FH detection linearly, assuming that the proportion of the population with FH identified will be low and that the number of FH cases identified twice is negligibly small.

## 2.2 Screening programme comparisons

Two systematic approaches to screening have been or are being clinically evaluated: (1) universal screening of 1-2-year-olds followed by reverse cascade screening of parents, and (2)

screening of electronic healthcare records. Here, we explore the efficacy and cost-effectiveness of both. In (1), universal screening is applied for a period of 10 years [34] and we apply a range of ratios of relatives-identified-to-index-cases, determined from published studies (see Table 2). In (2), we consider the outcomes of a feasibility study using the FH Case Ascertainment Tool 2 (FAMCAT2) [35], combined with a cascade screening programme, using the same ratios of relatives-identified-to-index-cases, determined from published studies.

The cost-utility of universal screening of 1-2-year-olds followed by cascade screening has been modelled. The most cost-effective strategy identified by ICER was cholesterol screening, followed by genetic testing and reverse cascade testing. This method is expected to identify 39.8 individuals with monogenic FH per 10,000 people screened (24.4 index cases, and 15.4 parents) at an ICER of £12,480 versus no screening [34]. The population of 1-2-year-olds in England is estimated to be 1,292,840 [36], meaning that 3,155 index cases and 1,991 parents can be expected to be identified if all 1-2-year-olds were screened. If all 1-2-year-olds in England were screened for a period of 10 years, we can expect approximately 15,773 index cases to be identified and 9,955 relatives [34]. The cost of screening was valued at £13,785 per diagnosis, and the cost of reverse cascade testing was valued at £1,110 per relative identified with FH [34].

The feasibility study of the FAMCAT2 algorithm for screening electronic health care records searched 193,589 GP records, identifying 86,219 eligible records that belonged to patients aged 18 years or older, with no previous diagnosis of FH and with a serum cholesterol measurement (total cholesterol or LDL-C) [34]. Of this group, 3,375 patients were invited to participate in the study, leading to 1,336 being recruited and ultimately 283 receiving genetic tests. This led to 11 true positive cases after a suitable FAMCAT2 detection threshold was chosen [35].

It would be reasonable to expect that, as part of a national screening programme, all eligible patients would be invited for screening. With the same proportions applied to all 86,219 patients eligible for screening, we would expect to identify 281 true positive FH cases per 193,589 records. The adult population of England aged 18 to 39 years is 16,922,140 [36]. We estimate that screening of all eligible adult healthcare records in this age-bracket would yield 24,564 index cases. In a follow-on from the same study, it was estimated that the incremental cost effectiveness ratio (ICER) for FAMCAT2 versus no active screening was £7,552 [37].

## 3. Results

### 3.1. FH identification in the UK falls well below FH identification in the Netherlands and Norway and is therefore unlikely to reach current targets

In 2011, the UK National Screening Committee (NSC) determined that an adult screening programme specific to FH was unnecessary and that the NHS Health Checks (HCs) programme should be adequate for identifying the majority of the population with FH [38]. The HCs commenced in 2009 for adults in England aged 40–74 years, with the aim of detecting signs of preventable conditions, including diabetes, heart disease, kidney disease, stroke, and dementia [39]. The HCs are intended to screen 3 million people per year, at a cost of £165 million per year, and to assess each of the eligible population of 15 million five-yearly [39].

However, by 2018, after nine years of the NHS Health Checks and fifteen years of genetic testing, it is estimated that only 3.98% of the population of England with FH had been identified [21], giving an estimate of the annual detection rate of 0.27% of the population of England with FH. This estimate includes the known population with FH prior to introduction of the testing and thus is an upper limit of the detection rate.

**Table 2. The profile of detection of national screening programmes.** Screening data comparing the proportions of service users (index cases and relatives) for Northern Ireland, the Netherlands and Norway, Wales, Scotland, Wessex (in pooled data) and the UK. *denotes estimated values for missing data based on the autosomal dominant nature of HeFH, where it affects 1 in 2 parents/children.

| Region | Study | Index cases identified | Relatives DNA tested | Relatives identified with FH after DNA testing | Relatives DNA tested per index case | Relatives identified with FH per relative DNA tested | Relatives identified with FH per index case |
|---|---|---|---|---|---|---|---|
| Northern Ireland | Kerr, 2017 | 254 | 1,736 | 841 | 6.83 | 0.48 | 3.31 |
| | Haralambos, 2018 | 296 | 1,955 | 960 | 6.60 | 0.49 | 3.24 |
| The Netherlands | Fouchier, 2005 | 2,818 | 14,158* | 7,079 | 5.02* | 0.5* | 2.51 |
| Norway | Leren & Bogsrud, 2021 | 2,829 | 14,230 | 5,993 | 5.03 | 0.42 | 2.12 |
| Wales | Haralambos, 2018 | 452 | 1,022 | 548 | 2.26 | 0.54 | 1.21 |
| United Kingdom | Haralambos, 2018 | 3,344 | 7,434 | 3,824 | 2.22 | 0.51 | 1.14 |
| England | Haralambos, 2018 | 1,603 | 2,782 | 1,484 | 1.74 | 0.53 | 0.93 |
| Scotland | Haralambos, 2018 | 993 | 1,675 | 832 | 1.69 | 0.50 | 0.84 |
| Scotland, Wales, and Wessex | Kerr, 2017 | 1,470 | 1,957 | 996 | 1.33 | 0.51 | 0.68 |

The LTP sets the target of identifying at least 25% of the population of England with FH by 2024 [15]. However, we can see in Fig 1 that this will be unsuccessful if detection continues at the upper estimate of 0.27% of cases per year. In fact, in this eventuality, the LTP target will only be reached in the year 2096. These estimates were based on data reported before the COVID-19 pandemic, where the pandemic is likely to have suppressed the detection rate to even lower levels.

In Fig 1, the most rapid growth in the identified populations can be seen for the Norway and Netherlands programmes, with detection since the start rates peaking at 2.50% and 2.07% of the FH population identified per year, respectively (see also Table 1). The fact that the more established and comprehensive Norway and the Netherlands programmes identify FH at rates below that required to satisfy the targets of the LTP implies that the LTP target was unrealistic from the outset.

## 3.2 Screening programmes are qualitatively different between nations

The proportion of index cases and relatives identified varies between screening programmes. Comparing national screening programmes, we can see that they show different profiles (see Fig 2). In particular, the England, Scotland and Wales programmes are less effective at identifying relatives as part of cascade screening than Norway and the Netherlands (see Table 2). Interestingly, Northern Ireland, as a constituent of the UK, independently follows a distribution much closer to Norway and the Netherlands with a higher proportion of relatives identified.

The UK identifies 1.14 FH positive relatives per index case, in contrast to Norway, the Netherlands and Northern Ireland alone who identify 2.12, 2.51, and 3.24 FH positive relatives per index case, respectively (see Table 2). If England had been able to increase the number of FH positive relatives identified per index case to 2.12 (Norway), 2.51 (Netherlands) or 3.24 (Northern Ireland) across the period of the LTP then by 2024 the proportion of the population with FH detected would be 7.76%, 8.24% or 9.14%, respectively (see Fig 3).

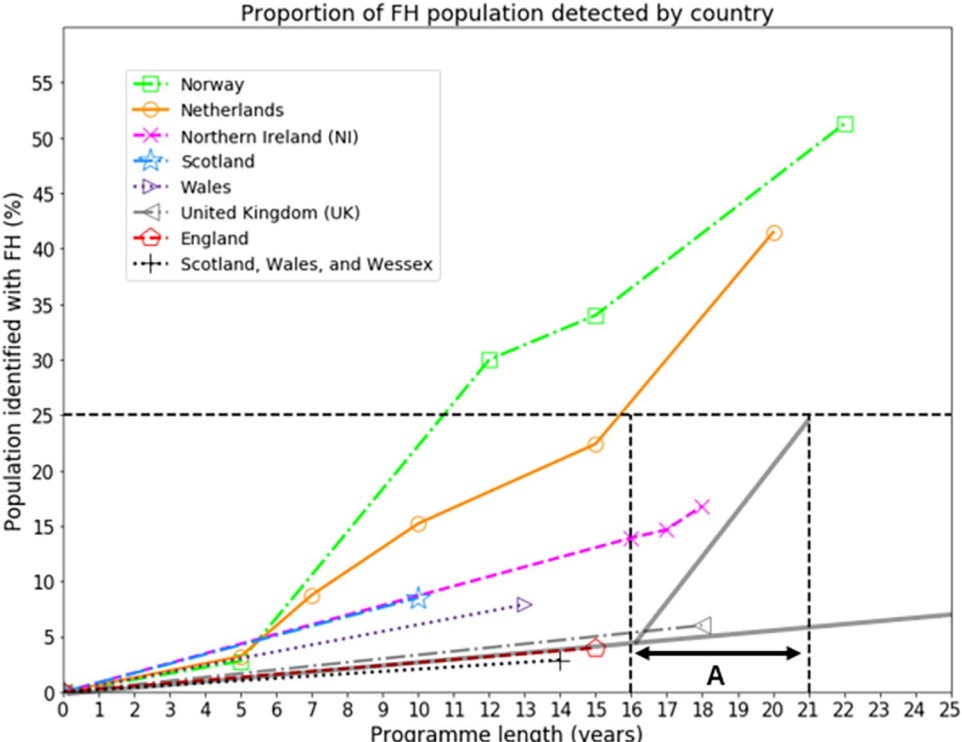

**Fig 1. Estimated percentage of national FH populations detected.** The estimated detection percentage of the national FH population is shown against time since genetic testing was available, for Norway (squares), the Netherlands (circles), the whole of the UK (left-pointing triangles), and constituents of the UK: England (pentagons, along with extrapolation in grey), Northern Ireland (crosses), Scotland (stars), Wales (right-pointing triangles), pooled data for Scotland, Wales, and Wessex (plusses). The period labelled 'A' represents the 5-year LTP target commencing in 2019 after 4.25% of the FH population in England were estimated to have been already identified. The trajectory of FH detection that would have been required through the period of the LTP (A) is also shown, equivalent to a detection rate of 4.15% per year, which is well above the peak detection rates of Norway and the Netherlands (see Table 1). Extrapolation determines that the LTP target will only be met after 77 years of the programme, in the year 2096.

Both the Netherlands and Norway have well-established national FH screening programmes. It is estimated that Norway has identified 51.33% of its FH population as of 2020, and that the Netherlands has identified 41.49% of its FH population as of 2014 (see Fig 4 and Table 1). In contrast, England identified 3.98% of its population from 2003 to 2018, Wales identified 7.89% of its population from 2005 to 2018, Scotland identified 8.5% of its population from 2003 to 2018, and Northern Ireland identified 16.88% of its population from 2000 to 2018. Pooled data for Scotland, Wales and the county of Wessex in England estimated that 2.88% of its combined FH population was detected from 2003 to 2017.

### 3.3 How could an efficacious and cost-effective national screening programme be introduced to England?

The NSC has recommended not to screen children for FH [39]. However, NHS England and NHS Improvement (NHSEI) will run an 18-month service evaluation of universal screening of 1-2-year-olds, provided by 3 to 5 of the 15 Academic Health Science Networks (AHSNs) across England [40]. This evaluation aims to screen 30,000 1–2-year-olds for FH at routine immunisation, commencing in 2021, and is expected to be in operation until 2023 [39, 41, 42]. A cost-utility analysis has shown that the most cost-effective molecular method by incremental cost-

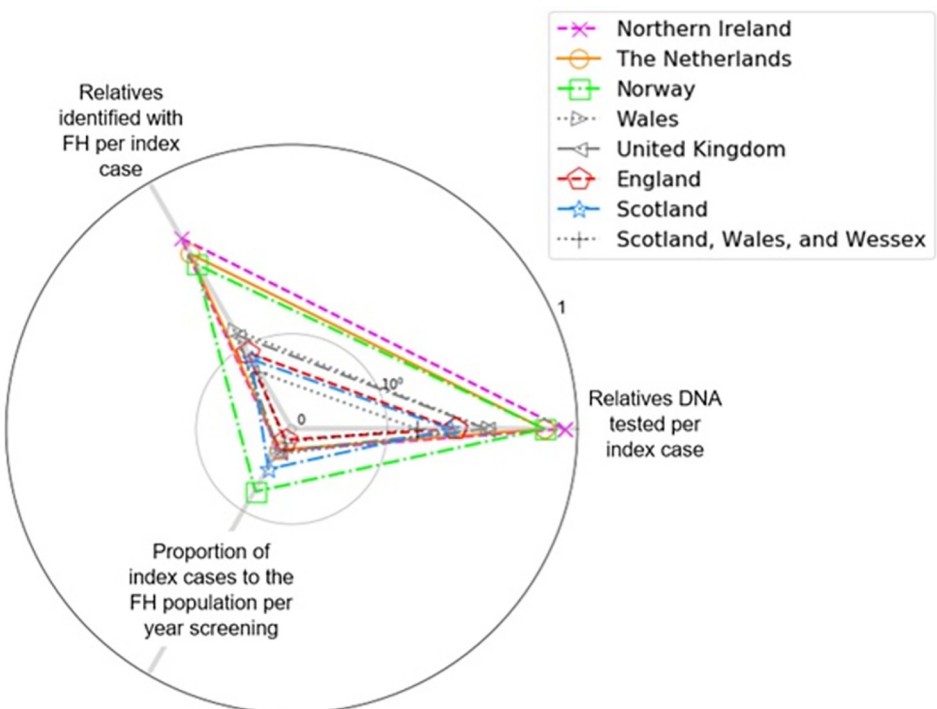

**Fig 2. Screening profile of national programmes.** Radar plot shown on a base-10 log scale. Screening in England, Scotland and Wales follows a different profile to Norway, the Netherlands and Northern Ireland (see Table 2). Norway, the Netherlands and Northern Ireland show a distinctly higher proportion of relatives tested per index case.

effectiveness ratio (ICER) per quality adjusted life year (QALY) was cholesterol screening, followed by genetic testing and reverse cascade testing, versus no screening [33]. The NSC also announced that systematic searching of adult healthcare GP records will be carried out. However, it has not been made clear the number of GP practices involved or records planned to be searched [40].

We modelled universal screening scaled to the age-appropriate population of England and coupled this to a reverse cascade screening programme that identified relatives at the rates observed for Norway, the Netherlands and the UK. The resulting proportion of FH cases identified is shown in Fig 5 along with the ICER. We also modelled screening of electronic health care records scaled to the age-appropriate population of England and coupled this to a cascade screening programme that identified relatives at the rates observed across countries and regions (also see Fig 5).

It is very clear that increasing the number of relatives identified per index case results in lower ICERs. Interestingly, electronic healthcare record screening for index cases results in a substantially lower ICER cost than universal screening. This gap is at its greatest when fewest relatives are screened per index case and becomes narrower when greater numbers of relatives are screened per index case, at which point the costs of index case detection, the most expensive stage, is spread over the greatest number of detected cases. At the lowest level of relative screening (0.68 relatives per index case) electronic health care record screening outperformed universal screening, reducing the ICER from £8,654.62 to £4,944.52 per case, a 43% reduction (see Table 3). At the greatest level of relative screening (3.24 relatives per index case), electronic health care screening reduced the ICER from £4,099.39 to £2,629.43 per case, a 36% reduction. Similarly, it is clear that electronic healthcare record screening for index cases is substantially

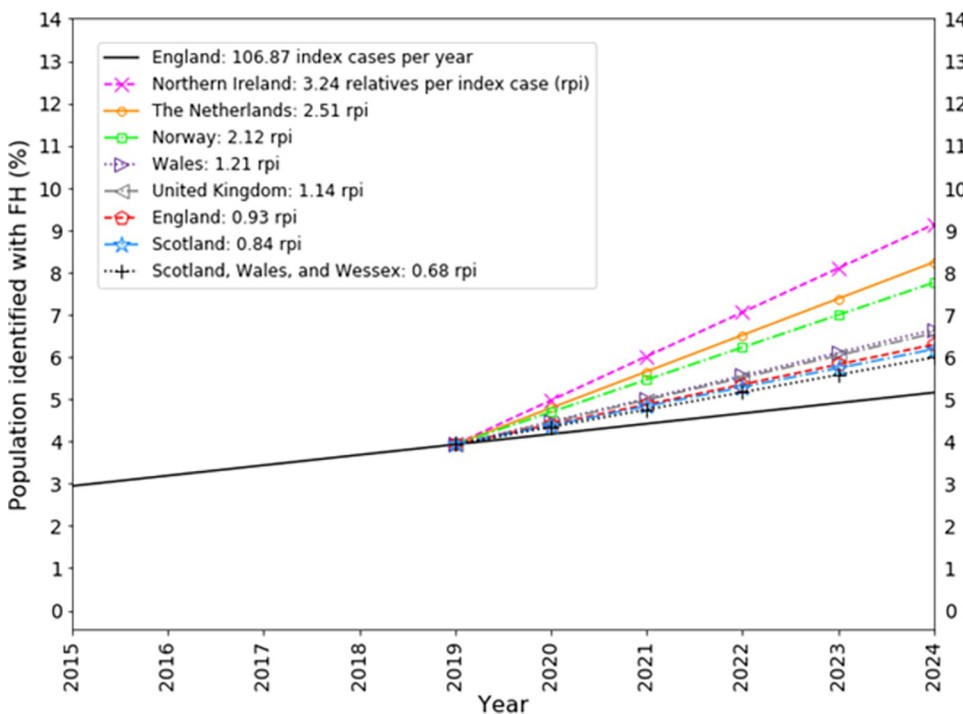

**Fig 3. The proportion of the UK population with FH identified across the 5 year period of the LTP after adopting alternative scenarios of FH positive relatives-identified per index case.** The FH population of England is identified at 0.27% per year (pentagons) and extrapolated across the period of the LTP. The proportion of the FH population of England that would be hypothetically identified across the period of the LTP is also shown if the number of relatives with FH identified per index case followed that of Northern Ireland (9.14%), The Netherlands (8.24%), Norway (7.76%), Wales (6.64%), United Kingdom (6.56%), England (6.29%), Scotland (6.18%), and Scotland + Wales + Wessex (5.99%).

more efficacious than universal screening and that this difference is amplified when greater numbers of relatives are screened per index case. With 0.68 relatives identified per index case, electronic health care record screening outperformed universal screening, increasing detection from 11.73% to 18.26%, a 56% increase. With 3.24 relatives identified per index case, electronic health care record screening outperformed universal screening, increasing detection from 29.60% to 46.09%, also a 56% increase.

## 4. Discussion

The NHS Long Term Plan contained a target to identify of 25% of the FH population of England across 5 years to 2024 and was published in January of 2019, well before the start of the COVID-19 pandemic. However, the UK's screening and detection rates can only have been reduced as a result of the additional demand placed on health care providers during the pandemic. Rates of detection are likely to continue to be suppressed as the backlog of post-pandemic care is addressed and as a result of reticence to travel by older and vulnerable patients, post-pandemic. The impact of this pandemic and of any future pandemics will have to be a consideration in any future screening targets. The impact of treatment backlogs on staffing and workloads provide further justification for the adoption of automated screening of electronic healthcare records, such as FAMCAT2 and TARB-Ex, which have been demonstrated to be effective without being labour intensive [34, 43, 44].

Monitoring progress of screening depends critically upon clear and comprehensive primary data. We have found that screening data has been reported to varying extents in different

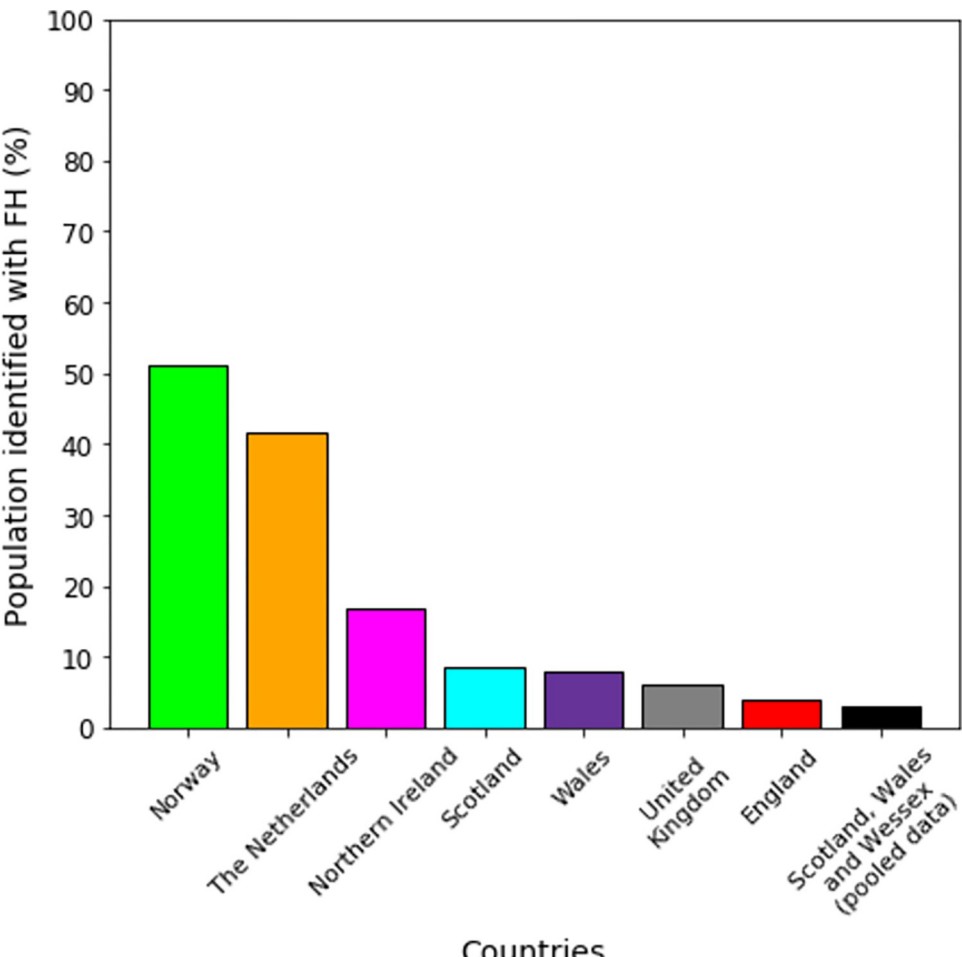

**Fig 4. The most recently reported percentage of the population with FH estimated to have been detected across countries/regions (see Table 1).**

jurisdictions. Five publications describe FH screening in the Netherlands, but only one describes screening for the whole of the UK (see Table 1). Furthermore, not all publications report the same data. To present the most complete picture possible, it was necessary for us to make estimates from primary data, though it is important to highlight that these estimates did not affect our key findings. This suggests that there should be a standard minimum data set to be reported for FH epidemiology studies:-

1. The number of index cases genetically identified

2. The number of relatives genetically tested

3. The number of relatives genetically identified with FH and VUS

In order to monitor the trajectory of screening programmes, this data would have to be reported periodically, ideally every two to three years.

It is clear that there is a significant mismatch between the target of 25% of the FH population in England and what is likely to be delivered within the timeframe of the LTP, with the target only likely to be reached in 2096 at current screening rates (see Fig 1 and Table 1). To achieve the targets set over the period of the LTP a detection rate of 4.15% per year would have

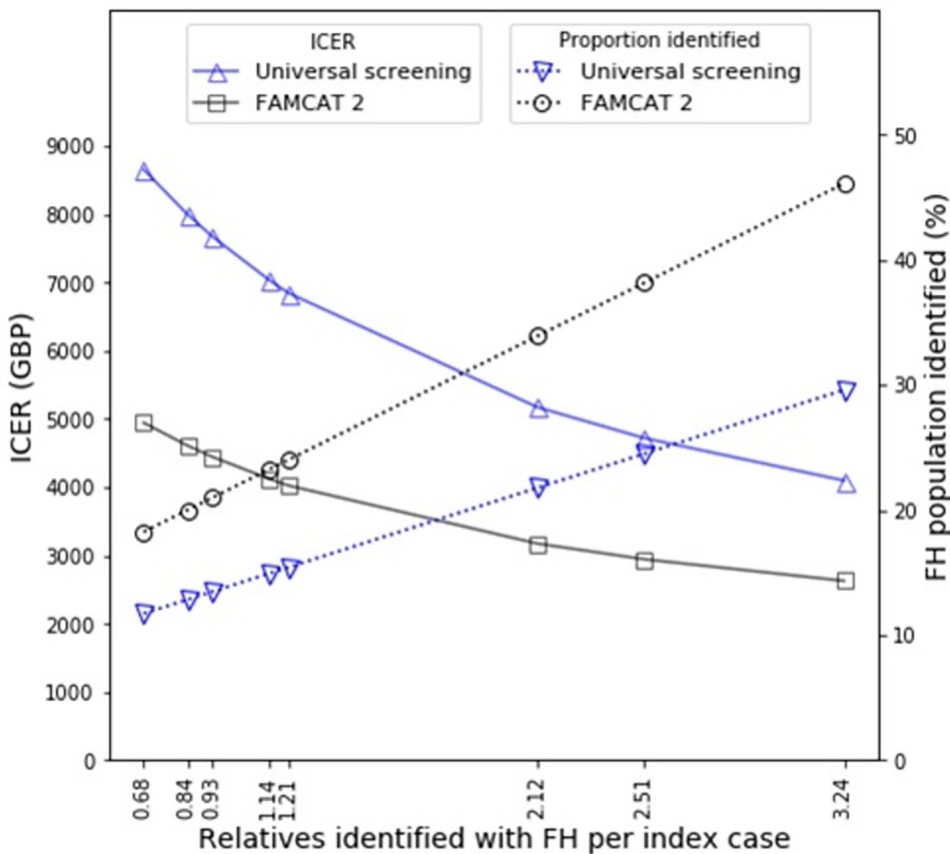

**Fig 5. Costs and efficacy for cascade screening following index case identification with either universal screening of 1-2-year-olds for 10 years or FAMCAT2 screening of electronic health care records for 18-39-year-olds.** The efficacy and cost-effectiveness is shown for cascade screening with a range of rates of FH positive relative identification per index case as determined from previous studies (see Table 2). ICERS are calculated per detected FH case detected versus no screening.

been required, exceeding that of both the Netherlands and Norway, countries with well-established national-scale programmes. The best performing period reported across any of the geographical regions was between 2003 and 2010 in Norway when 3.89% cases per year were identified. This suggests that the LTP targets were unrealistic from the outset and that any country aspiring to develop a national screening programme would either have to set more modest targets or invest more heavily in screening resources.

It is also clear that there is a difference in the profile of how detection occurs in Norway, the Netherlands and Northern Ireland versus England, Scotland and Wales (see Fig 2 and Table 2). The former group detected greater proportions of relatives per index case, presumably by placing a greater emphasis on cascade screening, which enabled them to achieve higher proportions of detection in their populations (see Fig 4 and Table 1) and to be able to use their genetic testing services more cost-effectively (see Fig 5 and Table 3).

When considering a range of screening programmes, Fig 5 would naively suggest that screening of electronic healthcare records will identify more index cases than universal screening of 1-2-year-olds to contribute to programmes of cascade screening. However, direct comparisons are challenging as our modelling of universal screening of 1-2-year-olds and FAMCAT2 detection are limited in different ways. Modelling of universal screening of 1-

**Table 3. Screening programme efficacy and cost.** We identified index cases with either universal screening of 1–2-year-olds for five years or FAMCAT2 screening of electronic healthcare records and applied cascade screening, using a number of relatives per index case determined from previous studies (see Table 2). It is clear that screening programmes that identify a higher proportion of relatives per index case are more efficacious and more cost-effective and that screening of electronic healthcare records outperforms universal screening.

| Relatives identified with FH per index case | Universal Screening of 1–2-year-olds for 10 years | | | | FAMCAT2 Screening (all adults 18 to 39 years) | | | |
|---|---|---|---|---|---|---|---|---|
| | Total population with FH identified | Proportion identified | Cost | Cost/Person (ICER) GBP | Total population with FH identified | Proportion identified | Cost | Cost/Person (ICER) GBP |
| 3.24 (Northern Ireland) | 66,876.03 | 29.60% | £274,150,703.95 | 4,099.39 | 104,147.12 | 46.09% | £273,838,149.2 | 2,629.34 |
| 2.51 (Netherlands) | 55,361.99 | 24.50% | £261,370,127.27 | 4,721.11 | 86,216.13 | 38.16% | £253,934,750.3 | 2,945.33 |
| 2.12 (Norway) | 49,210.66 | 21.78% | £254,542,147.95 | 5,172.50 | 76,647.22 | 33.92% | £243,301,427.6 | 3,174.74 |
| 1.21 (Wales) | 34,857.55 | 15.43% | £238,610,196.21 | 6,845.29 | 54,284.23 | 24.02% | £218,490,341.3 | 4,024.93 |
| 1.14 (United Kingdom) | 33,753.47 | 14.94% | £237,384,661.46 | 7,032.90 | 52,564.82 | 23.26% | £216,581,796.2 | 4,120.28 |
| 0.93 (England) | 30,441.21 | 13.47% | £233,708,057.21 | 7,677.36 | 47,406.59 | 20.98% | £210,856,160.9 | 4,447.82 |
| 0.84 (Scotland) | 29,021.67 | 12.84% | £232,132,369.68 | 7,998.59 | 45,195.92 | 20.00% | £208,402,317.2 | 4,611.09 |
| 0.68 (Sco, Wal and Wessex) | 26,498.05 | 11.73% | £229,331,147.39 | 8,654.64 | 41,265.84 | 18.26% | £204,039,928.4 | 4,944.52 |

2-year-olds has been limited to 10 years, but to comprehensively detect a high proportion of the FH population of England would require universal screening of an entire generation, i.e., 25–30 years. Similarly, our FAMCAT2 analysis was limited to the 18-39 age range of the population of England, rather than the whole population. The principal reason for limiting the range of the analysis in this way was that our modelling was linear and therefore well adapted to programmes that identify low and growing proportions of the FH population where there is only a small probability of a FH case being identified twice and overcounted. However, as the proportions grow large and approach a significant fraction of the entire population, alternative non-linear modelling methods are required that explicitly avoid overcounting of cases. Given that limiting screening was necessary to ensure accuracy, in the FAMCAT2 case, we felt that an earlier age range was the best choice due to the higher standardised mortality ratio indicating the greater role of elevated cholesterol as a risk factor [8, 9].

It is more straightforward to compare the incremental cost effectiveness ratio versus no intervention for cascade screening with either the universal screening of 1-2-year-olds or FAMCAT2. From Fig 5, we can see that FAMCAT2 screening offers significantly greater value for money per FH case detected. At the lowest proportion of relatives identified per index case (0.68), each FH detection costs 43% less using FAMCAT2 than universal screening (see Table 3). This difference reduces in magnitude as cascade screening becomes more effective and with 3.24 relatives identified per index case, using FAMCAT2 for index case detection reduces the cost per detection by 36% over universal screening. In real world application, the justification for using FAMCT2 screening is likely to be even greater as we start to consider factors such as the greater ease of gaining patient consent, the speed with which an electronic screen of healthcare records can be undertaken when compared to a universal screen and that parents typically have their children within a relatively short number of years, increasing the chances of double identification in regional screens (once by universal screening and again by cascade screening from a sibling) which reduces efficiency.

The NSC recommendation that the Heath Checks programme should serve as screening for the FH population raises a number of concerns. The NSC has responsibility for advising screening throughout the UK. However, both the LTP target and the NHS HCs are specifically programmes of NHS England, with no equivalent in place for NHS Scotland, Wales, or

Northern Ireland. Hence, the decision creates a two-tier standard of care based on geography. A further concern is the low uptake of HCs by the population of England. Uptake has been shown to vary regionally from 25.1% to 84.7% and, in a review of 9.7m patients it was found that only 52.6% had taken up the invitation [45]. By channelling FH screening through the HCs, the LTP target of identifying 25% of the FH population of England actually corresponds to identifying 48% of the FH population of England who are accessible through HC screening (i.e., the population of England who accept the HC invitation and their relatives), which in practice is a highly challenging target. A further concern lies in the age at which screening occurs and the likely impact on quality of life. Screening programmes that target earlier ages support interventions with lifelong benefit. However, screening programmes that target adults in later life only enable the initiation of treatment at an older age by which time significant morbidity may have developed. Hence, early life screening is likely to have greater impact on patient health.

## 5. Conclusion

We have shown that the profile of screening programmes varies between countries and regions and that Norway, the Netherlands and Northern Ireland all have significantly more effective cascade screening than England, Scotland and Wales. As a result, Norway, the Netherlands, and Northern Ireland have detected a higher proportion of their FH populations than England, Scotland and Wales and at an estimated lower cost per detection. We have also explored the relative value of identifying index cases by screening electronic healthcare records versus universal screening of 1–2-year-olds, showing that electronic healthcare record screening is significantly less expensive per case detected with costs 36–43% lower. These results suggest that electronic healthcare record screening combined with effective cascade screening would offer the greatest efficacy and cost-effectiveness.

The target to identify 25% of the population of England within the 5 years from 2019 to 2024 does not appear to be realistic. Reaching this target would require out-performing well-established national programmes that are recognised as world-leading in FH detection. At the current rate of detection, we have shown that this target will only be reached for the FH population of England in 2096. For NHS England and the UK more widely, electronic healthcare record screening combined with effective cascade screening would be the best strategy to follow in pursuit of case detection targets for familial hypercholesterolemia.

## Author Contributions

**Conceptualization:** Christopher Page, Maurice O'Kane, Steven Watterson.

**Data curation:** Christopher Page.

**Formal analysis:** Christopher Page.

**Funding acquisition:** Taranjit Singh Rai, Steven Watterson.

**Investigation:** Christopher Page.

**Methodology:** Christopher Page, Huiru Zheng, Haiying Wang, Taranjit Singh Rai, Steven Watterson.

**Project administration:** Christopher Page, Huiru Zheng, Haiying Wang, Taranjit Singh Rai, Steven Watterson.

**Software:** Christopher Page.

**Supervision:** Huiru Zheng, Haiying Wang, Taranjit Singh Rai, Pádraig Hart, Shane McKee, Steven Watterson.

**Visualization:** Christopher Page.

**Writing – original draft:** Christopher Page, Steven Watterson.

**Writing – review & editing:** Christopher Page, Huiru Zheng, Haiying Wang, Taranjit Singh Rai, Maurice O'Kane, Pádraig Hart, Shane McKee, Steven Watterson.

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
