## [Decision Letter · Decision Letter 0]

20 Feb 2023

PGPH-D-22-01978

A comparison of the Netherlands, Norway, and the UK Familial Hypercholesterolemia screening programmes with implications for target setting and the UK’s NHS Long Term Plan

Dear Dr. Watterson,

Thank you for submitting your manuscript to PLOS Global Public Health. After careful consideration, we feel that it has merit but does not fully meet PLOS Global Public Health’s publication criteria as it currently stands. Therefore, we invite you to submit a revised version of the manuscript that addresses the points raised during the review process.

EDITOR: Please insert comments here and delete this placeholder text when finished. Be sure to:

Indicate which changes you require for acceptance versus which changes you recommendAddress any conflicts between the reviews so that it's clear which advice the authors should followProvide specific feedback from your evaluation of the manuscript

Please ensure that your decision is justified on PLOS Global Public Health’s publication criteria and not, for example, on novelty or perceived impact.

We look forward to receiving your revised manuscript.

Kind regards,

Panniyammakal Jeemon

Academic Editor

Journal Requirements:

Additional Editor Comments (if provided):

Reviewers' comments:

Reviewer's Responses to Questions

**Comments to the Author**

1. Does this manuscript meet PLOS Global Public Health’s publication criteria? Is the manuscript technically sound, and do the data support the conclusions? The manuscript must describe methodologically and ethically rigorous research with conclusions that are appropriately drawn based on the data presented.

Reviewer #1: Partly

Reviewer #2: Yes

2. Has the statistical analysis been performed appropriately and rigorously?

Reviewer #1: N/A

Reviewer #2: I don't know

3. Have the authors made all data underlying the findings in their manuscript fully available (please refer to the Data Availability Statement at the start of the manuscript PDF file)?

Reviewer #1: Yes

Reviewer #2: Yes

4. Is the manuscript presented in an intelligible fashion and written in standard English?

Reviewer #1: Yes

Reviewer #2: Yes

5. Review Comments to the Author

Reviewer #1: The article dealt with an issue of timely relevance and put forward sound data-backed conclusions.

However I have a few questions on the article which I am stating below.

1.Authors have chosen Norway and Netherlands as countries for comparison. I would welcome some references which establish that these are the countries with well established screening programs, which is the reason authors have given for selecting these particular countries.

2.In Page 11 line 223 its said that number of FH positive relatives identified per index case is 2.12., 2.52 and 3.24 for Norway , Netherlands and Northern Ireland respectively. The same values are also given in Table 2. However, in line 225 the values mentioned for FH positive relatives identified per index case is 2.74(Norway), 3.12(Netherlands) and 3.24(Northern Ireland). Why is there a (slight) discrepancy and what is the source of the latter set of values?

3.In Table 2, Relatives identified with FH per index case is not equal to the ratio of Relatives identified with FH after DNA testing and Index cases identified; for United Kingdom. The value given in the table is 0.99 but the value calculated from the latter two columns comes to 1.12.

4.In line 186, it is mentioned that The LTP sets the target of identifying at least 25% of the FH population of England by 2024 . However, we can see in Figure 4 that this will be unsuccessful if detection continues at the upper estimate of 0.27% of cases per year.

However, Figure 4 just shows proportion of FH cases identified without making explicit the rate of detection or time period.

Reviewer #2: The paper is very readable and is of interest in formulating a national screening program. Screening for Familial hypercholesterolemia is now a Class I recommendation and in this context comparison of well established screening programs is commendable

6. PLOS authors have the option to publish the peer review history of their article (what does this mean?). If published, this will include your full peer review and any attached files.

**Do you want your identity to be public for this peer review?** For information about this choice, including consent withdrawal, please see our Privacy Policy.

Reviewer #1: No

Reviewer #2: No

---

## [Editor Report · Decision Letter 1]

29 Mar 2023

A comparison of the Netherlands, Norway, and UK Familial Hypercholesterolemia screening programmes with implications for target setting and the UK’s NHS Long Term Plan

PGPH-D-22-01978R1

Dear Dr Watterson,

We are pleased to inform you that your manuscript 'A comparison of the Netherlands, Norway, and UK Familial Hypercholesterolemia screening programmes with implications for target setting and the UK’s NHS Long Term Plan' has been provisionally accepted for publication in PLOS Global Public Health.

Best regards,

Panniyammakal Jeemon

Academic Editor